# Mapping and Quantifying Comprehensive Land Degradation Status Using Spatial Multicriteria Evaluation Technique in the Headwaters Area of Upper Blue Nile River

**Alelgn Ewunetu [1,2,\*], Belay Simane [2], Ermias Teferi [2,3] and Benjamin F. Zaitchik [4]**

[1] Department of Geography and Environmental Studies, Woldia University, Woldia 400, Ethiopia
[2] Center for Environment and Development Studies, Addis Ababa University, Addis Ababa 1176, Ethiopia; belay.simane@aau.edu.et (B.S.); ermias.teferi@aau.edu.et (E.T.)
[3] Water and Land Resource Center, Addis Ababa University, Addis Ababa 1176, Ethiopia
[4] Department of Earth and Planetary Sciences, Johns Hopkins University, Baltimore, MD 21218, USA; zaitchik@jhu.edu
[\*] Correspondence: alelgn.ewuntu@aau.edu.et; Tel.: +251-912-772-560

**Abstract:** Mapping and quantifying land degradation status is important for identifying vulnerable areas and to design sustainable landscape management. This study maps and quantifies land degradation status in the north Gojjam sub-basin of the Upper Blue Nile River (Abbay) using GIS and remote sensing integrated with multicriteria analysis (MCA). This is accomplished using a combination of biological, physical, and chemical land degradation indicators to generate a comprehensive land degradation assessment. All indicators were standardized and weighted using analytical hierarchy and pairwise comparison techniques. About 45.3% of the sub-basin was found to experience high to very high soil loss risk, with an average soil loss of 46 t ha$^{-1}$yr$^{-1}$. More than half of the sub-basin was found to experience moderate to high level of biological degradation (low vegetation status and low soil organic matter level). In total, 80.2% of the area is characterized as having a moderate level of physical land degradation. Similarly, the status of chemical degradation for about 55.8% and 39% of the sub-basin was grouped as low and moderate, respectively. The combined spatial MCA of biological, chemical, and physical land degradation indicators showed that about 1.14%, 32%, 35.4%, and 30.5% of the sub-basin exhibited very low, low, moderate, and high degradation level, respectively. This study has concluded that soil erosion and high level of biological degradation are the most important indicators of land degradation in the north Gojjam sub-basin. Hence, the study suggests the need for integrated land management practices to reduce land degradation, enhance the soil organic matter content, and increase the vegetation cover in the sub-basin.

**Keywords:** comprehensive land degradation; soil loss; GIS; MCA; Upper Blue Nile

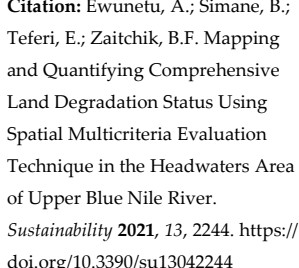

## 1. Introduction

Land is a critical natural resource for human survival and the base for all terrestrial ecosystem services. Healthy land qualities are vital for sustainable development, including food security and improved livelihoods [1], serving as a key factor in many production processes and economic growth [2,3]. Unsustainable use of land resources, however, results in degradation of land quality and quantity [1]. Land degradation here primarily refers to the loss of life-supporting land resources through a mix of processes that include soil erosion, soil compaction, destruction of soil structure, loss of soil organic matter, loss of vegetation cover, desertification, salinization, and acidification [4,5]. Land degradation reduces the economic value of ecosystem services and goods derived from land resource bases [6]. It has become a problem of great concern due to its impacts on food production, water supply, energy supply, and ecosystem services as a whole. According to the United

Nations Convention to Combat Desertification (UNCCD), about 25% of the total land area of our planet Earth is severely degraded or undergoing degradation [7]. Africa is the continent most severely affected by land degradation [8,9]. According to the estimates made by the UNCCD [10], land degradation affects up to two thirds of productive land area in Africa. In Ethiopia specifically, land degradation is a chronic and ongoing problem [11–13]. More than 85% of country's landmass is degraded to some degree [14] due to population pressure, persistent poverty, rugged topography, climatic condition, use of biomass for fuel, lack of awareness, and misuse of land management technologies [11,12,14–16].

Deforestation is a continuing process of land degradation and results in local soil erosion, biodiversity loss, changes to hydrology and climate, and contributes to global climate change [17]. Forest covered about 40% of Ethiopia's land area at the beginning of the 20th century and reduced to 2.36% cover in 2000 [18]; however, recent evidence show that forest cover has rebounded to 12% [19]. Soil erosion is a highly visible form of land degradation in Ethiopia that has depleted topsoil for many years, primarily through water erosion [20–23]. For example, it is estimated that northern Ethiopia has lost 45 t ha$^{-1}$yr$^{-1}$ topsoil due to water erosion alone [24]. Estimates from northwestern Ethiopia indicate topsoil loss rates of 33.7 t ha$^{-1}$yr$^{-1}$ [25]. Similarly, on average, about 27.5 t ha$^{-1}$yr$^{-1}$ topsoil resource is being eroded from the entire Upper Blue Nile (Abbay) basin [26]. In addition, soil acidity has become a serious concern in the Ethiopian highlands in recent decades [27]. The study area, North Gojjam sub-basin, is in the Ethiopian highlands and was historically well known to be an area of high potential for agricultural production [13], but land resources of the area have been continuously degraded and the ecosystem productivity has declined at an alarming rate [28,29].

Mapping and quantifying land degradation status plays an important role in cost-effective design of land management strategies. It enables researchers and land managers to identify the most vulnerable areas and to give priority of the locational intervention [30,31]. Evidence of land degradation assessment can be collected using myriad methods, such as questionnaires, focus group discussions, direct observation, and expert judgment [32]. The application of remote sensing technologies has provided great opportunities to provide evidence of degradation over large areas at relatively low cost and time investment [32,33]. The indicators of land quality degradation assessment include physical, biological and chemical factors, as well as socio-economic changes [34–36].

Previous studies on mapping and quantifying land degradation have focused on the most visible forms of degradation, such as soil erosion and deforestation [26,30]. In reality, a range of indicators of land degradation, including soil fertility status, soil acidity, salinization, and loss of vegetation cover also have to be considered to assess the level of land degradation [37]. This multifactorial approach is necessary because the design of appropriate land management for sustainable development requires a holistic inventory and rating of vulnerable areas for degradation [35,38] according to multiple risks and potential interventions. It is also important to update assessment regularly, as land degradation is a dynamic process that responds to coupled natural-human factors [39].

It can be challenging to integrate diverse degradation metrics that are obtained through different measurement techniques and are not always directly comparable to each other. Spatial multi-criteria analysis (MCA) can be of considerable use in land degradation analysis. MCA methods are widely used for problems that involve multiple factors and multiple perspectives. In this study, a new approach for development of a comprehensive land degradation map is proposed, in which remote sensing, GIS, analytical hierarchy process (AHP), and MCA techniques are applied to account for many factors responsible for land degradation. Specifically, the study aims to: (1) estimate average annual soil loss; (2) map and quantify key biological, physical, and chemical land degradation status; and (3) develop comprehensive land degradation indicators.

This study provides empirical evidence to policy makers and land users to identify specific areas vulnerable to land degradation as well as offering a comprehensive assessment of land degradation vulnerability the north Gojjam sub-basin. This information is

valuable for informing environmental rehabilitation activities and sustainable agriculture. The rest of the paper is organized as follows: section two gives a brief overview of the study area and MCA procedures, along with a description of data; results are presented in section three; section four concludes with the main findings of the paper.

## 2. Materials

### 2.1. Description of the Study Area

The North Gojjam sub-basin is located between 38.2° E to 39.6° E longitude and 10.8°N to 11.9° N latitude. It is one of the major tributaries of the Blue Nile/Abbay river (Figure 1). Its area covers about 1,431,360 ha, which stretches between Choke and Guna mountains. The total population of the sub-basin and surrounding villages at the time of the study was 3,565,892 [40], settled in scattered areas. The altitude of the sub-basin ranges from 1044 to 4048 masl (Figure 1). The dominant agro-ecological zone is characterized by tepid to cool moist middle highlands, and cold to very cold moist sub afro-alpine to afro-alpine highlands [41]. According to the Ethiopian National Metrological Agency, the average maximum and minimum temperature of the sub-basin varies from 24.6–28.1 °C and 11.0–14.5 °C, respectively, and the mean annual temperature is 19.4 °C [42]. The rainfall pattern is closely correlated with the annual migration of the inter-tropical convergence zone (ITCZ) and most rainfall occurs in the summer, from June to September [41,42].

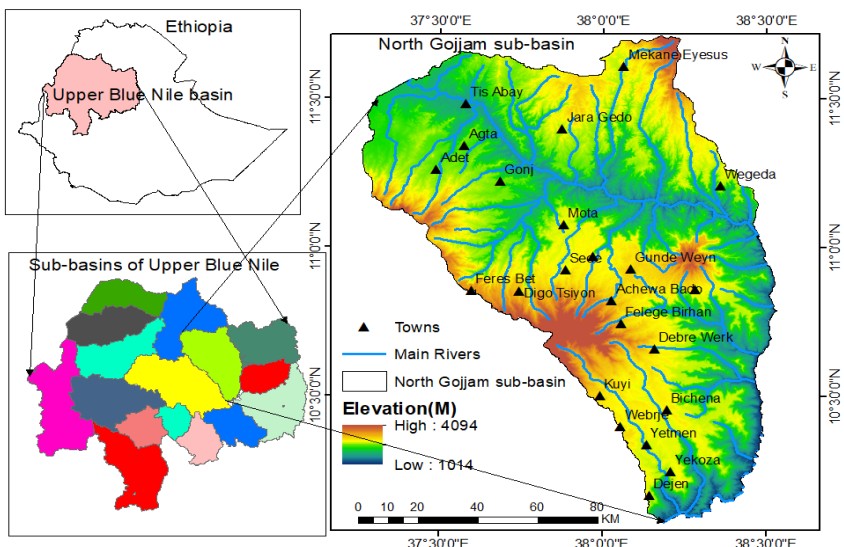

**Figure 1.** Location of North Gojjam sub-basin and its topography.

Meteorological records from stations within the sub-basin and in the surrounding area indicated that the mean annual rainfall (1986–2017) is 1334.48 mm with a minimum of 810 mm and a maximum of 1815 mm [42]. The dominant soil types are leptosols, vertisols, luvisols, and alisols [43]. The geology of the sub-basin is mainly dominated by basalt, but the lowlands are dominated by sandstone [43]. Natural forest cover is very low and found on riverbanks, hillsides, and in church yards, to some extent. *Eucalyptus globules* plantation is dominant among introduced trees, particularly in the highland regions. Unreliable rain-fed mixed crop and livestock production agriculture is the primary source of livelihood for the majority of the population in the sub-basin. Various cereal, pulse, and oilseed crops are widespread, with vegetable and fruit crops produced to some extent. Livestock types include cattle, sheep, goats, horses, donkeys, mules, and poultry.

### 2.2. Data Sources

Landsat images were downloaded from the US Geological Survey (USGS) Earth Explorer [44] for the month of January, in the dry season, to obtain low cloud cover images. ASTER Global Digital Elevation Model (ASTER-GDEM) data were obtained from [45]. Currently, precipitation estimations based on satellite products have developed an alternative source of sparse rainfall gauge data for different hydrologic practices, mainly in limited data regions like Africa which suffer from a scarcity of surface monitoring resources. Among others, Climate Hazards Group InfraRed Precipitation with Station data (CHIRPS) is commonly used in research [46]. For the present study, rainfall data from the years 2009–2017 were obtained from the CHIRPS website [47]. Gridded soil data with 250 m spatial resolution were downloaded from the African Soil Information Service (AFSIS) website [48]. A total of 2500 ground truth data points for land use and land cover (LULC) classification and accuracy assessment were collected from the field based on in-situ field observation, with geolocation obtained using a handheld Garmin GPS instrument and from Google Earth using a time slider image [49]. Of these total ground truth points, 738 were used for accuracy assessment of image classification using the error matrix, which allowed us to evaluate the kappa coefficient, overall accuracy, and the producer's and user's accuracy. To capture additional information, including qualitative perceptions, 18 focus group discussions (FGDs) were carried out in 9 villages with 127 (78 male and 47 female) community members selected from upper, middle, and lower parts of the sub-basin, and interviews were administered for 27 farmers, 9 Development Agents (DAs), and 9 districts' crop, livestock, and natural resource experts to obtain additional information regarding the study area.

### 2.3. Spatial Multi-Criteria Analysis (MCA)

Evaluating comprehensive land degradation is not a trivial task, because it includes a wide range of factors that can be challenging to combine into a single index. To address this challenge, MCA can be applied to a combination of geospatial datasets [50]. MCA is an evaluation technique that is used to identify vulnerable areas for sustainable natural resource management [51–53] by ranking or scoring the performance of decision options against multiple criteria [54]. The MCA process includes several steps: describe objectives, select the criteria to measure the objectives, identify alternatives, renovate the criterion scales into commensurable units, assign weights to the criteria that reveal their relative importance, choose and apply a mathematical algorithm for ranking alternatives, choose an alternative, and combine criteria into a single index [51–56]. Spatial MCA operates as a raster process on multiple digital input maps of relevance to land quality indicators.

Based on the nature of the alternatives to be evaluated, MCA methods can be classified in two major types: continuous and discrete [57]. Continuous methods are used to determine an optimal quantity, which can vary continuously in a decision problem [52]. Discrete methods can be defined as decision support techniques that have a finite number of alternatives, a set of objectives and criteria by which the alternatives are to be judged, and a method of ranking alternatives depending on how well they satisfy the objectives and criteria [58]. Discrete methods can be further subdivided into weighting and ranking methods. These categories can be additionally subdivided into qualitative, quantitative, and mixed methods. Qualitative methods use only ordinal performance measures, while mixed qualitative and quantitative methods use different decision rules based on value and utility using mathematical functions [58]. The present study applied the discrete method because the indicators are finite and categorized.

There is a wide range of decision-making techniques in the MCA literature that are used to model complex problems [59]. The most common include: analytic hierarchy process (AHP) [58,60]; technique for order of preference by similarity to ideal solution (TOP-SIS) [61]; preference ranking organization method for enrichment evaluations (PROME-

THEE) [62]; complex proportional assessment method (COPRAS) [63]; additive ratio assessment (ARAS); visekriterijumska optimizacija I kompromisno resenje (VIKOR) method [64,65]; multi-attribute utility theory (MAUT) [66]; and others. However, these methods have been criticized due to an issue called the rank reversal problem (RRP). RR refers to a change in the ordering among alternatives previously defined, after the addition or removal of an alternative from the group previously ordered [56]. The characteristic objects method (COMET) is completely free of the rank reversal paradox and can be used in exchange for the AHP method [56], but the most popular method in the field of natural resource management is the AHP method. The theoretical foundations of AHP were developed by Saaty [58]. It is a prominent and powerful tool for making decisions in conditions relating multiple objectives. The present study used the AHP method integrated with GIS. The integration of GIS and AHP is a powerful approach to identify the suitable areas for agriculture and vulnerable areas to land degradation [51,57]. The indicators of the study are not complex and, thus, are not subjected to the rank reversal problem as we analyzed manually.

Here we implement MCA in combination with an AHP that allows pairwise comparison and applies weights to each factor when merging to a single output index [52]. Three hierarchical levels were applied (Figure 2). In the first hierarchy, individual land degradation indicator raster maps were prepared based on international standard values which were developed by organizations and scientists. In the second hierarchical level, primary indicators are grouped into classes. Vegetation cover [67] and soil organic matter [68,69] were grouped as biological land degradation indicator alternative. Soil erosion [70], soil compaction [71], soil drainage [72], and soil depth [73] were grouped as physical land degradation indicator alternatives. Soil acidity represented a single chemical land degradation indicator [69]. These groupings yielded a biological degradation index, physical degradation index, and chemical degradation index alternatives. In the final hierarchical stage, these three grouped indicators are combined into a single inclusive land degradation index (Figure 2).

MCA requires that the indicators be adjusted into similar units and a common value scale to make the comparison meaningful [52]. Value scaling, or standardization, is a process of converting different indicators to a common unit and scale. In doing so, it is possible to generate comparable alternatives according to their perceived relevance [74]. Linear scale transformation is the most commonly used method for criteria standardization in spatial MCA [75], and it is applied here for ease of interpretation. All criteria were standardized and then scaled to a value range from 1 to 5, representing very low to very high degradation level. There are three common weighting methods in spatial MCA: ranking, rating, and pairwise comparison [76,77]. Of these, the pairwise comparison method is most widely used in spatial MCA to estimate the overall weight of individual criteria [60], and we apply this method here. The criteria were weighted through the pairwise comparison of individual land degradation indicators which derived from raster data (maps) following AHP approach. In this case, when two criteria are compared, the less important criterion gets a reciprocal value of the most important. We use the principal eigenvalue and the corresponding normalized right eigenvector of the comparison matrix to provide the relative importance of the criteria being compared. The elements of the normalized eigenvector were weighted with respect to the criteria or sub-criteria and rated with respect to the alternatives [60,77]. The consistency of the matrix of order was then evaluated. If the consistency index failed to reach a threshold level, the comparisons were re-examined. Finally, the weighted overlay technique was used to combine/aggregate the criteria maps, and each standardized criterion was multiplied by its weight in the overlay process. All geospatial procedures are implemented using ArcGIS10.5 [78].

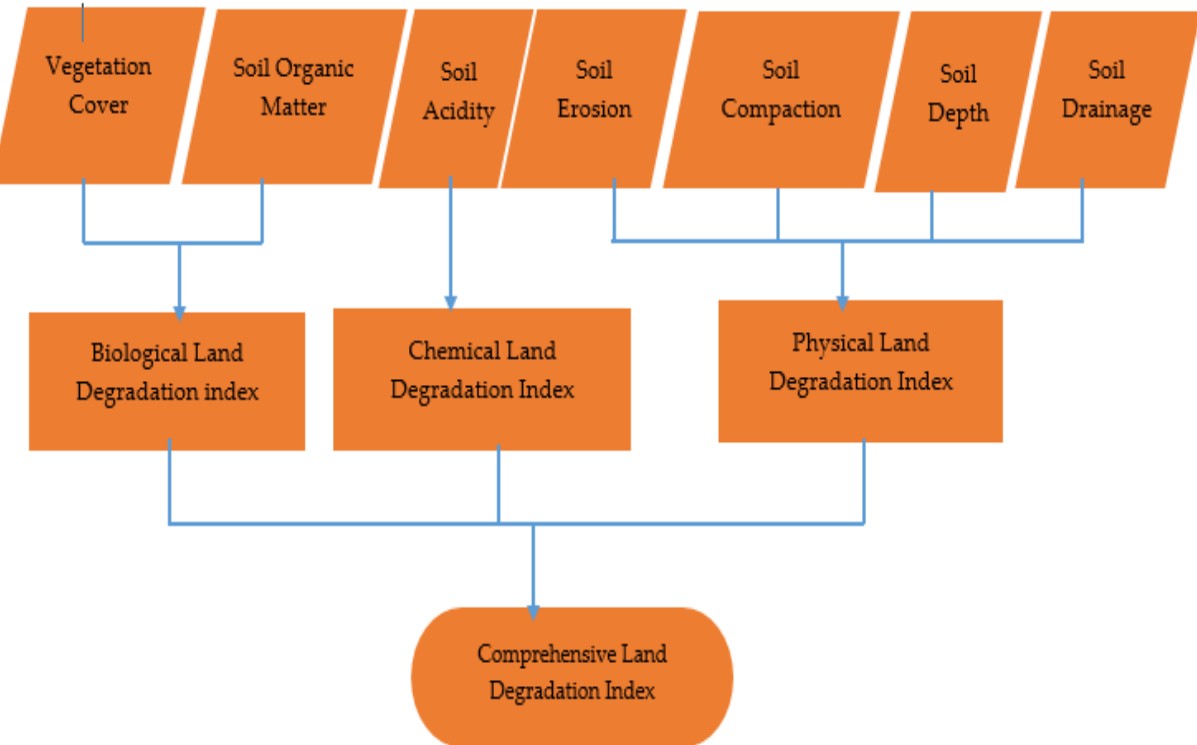

**Figure 2.** Hierarchical structures of land degradation status applied in multi-criteria analysis (MCA).

### 2.3.1. Develop Physical Land Degradation Indicators

A wide range of indicators can be applied to characterize physical soil/land degradation and the vulnerability of ecosystem health. Among others, soil compaction, soil drainage, and soil depth are commonly applied as indicators of soil ecosystem service productivity and, in turn, land degradation status [70–73]. We considered soil erosion, soil compaction, soil drainage, and soil depth as physical land degradation indicators.

#### Soil Loss Estimation Using RUSLE Model

Soil erosion is the main indicator of land degradation [11,12,79]. The agent of soil erosion may be water and/or wind. Given the humid climate condition of large parts of Ethiopian highlands, including the study area, water erosion is the dominant form of soil erosion. Soil loss by water can be estimated using the revised universal soil loss equation (RUSLE) [80]. The RUSLE is a very powerful tool when integrated with GIS, especially for developing countries that lack data for more physically based models. Some of the other reasons for the selection of the model for this study include its simplicity to apply and its compatibility with remote sensing data and GIS inputs in a computer interface [81]. Most of the input parameters of the model were adjusted for the Ethiopian highland context in a previous study [82]. The model estimates soil loss by taking into consideration rainfall, soil properties, topography, cover management, and conservation practices in a particular area [83].

$$A = R. K. LS. C. P \qquad (1)$$

where "A" is the mean annual soil loss (t ha$^{-1}$yr$^{-1}$); "R" represents a rainfall erosivity factor (MJ mm ha$^{-1}$h$^{-1}$yr$^{-1}$); "K" is a soil erodibility factor (tha$^{-1}$MJ$^{-1}$mm$^{-1}$); "LS" is a dimensionless topographic factor that accounts for, length of the slope (L) and slope steepness (S). "C" describes land cover factor and "P" is the erosion control support practice factor.

Rainfall erosivity factor (R-factor) indicates the input, which determines the sheet and rill erosion processes. R-factor depends mainly on the amount, intensity, and distribution of rainfall [81]. Soil loss is strongly associated with rainfall amount, intensity, energy, duration, pattern, and size of a raindrop through the detaching power of a raindrop striking the soil surface and for the incidence of runoff on the soil surface [83]. However, such data do not exist for the study area, due to the absence of an automatic rain-gauge, so, R-factor was generated using long-term mean annual rainfall following the equation developed by Hurni [82].

$$R = -8.12 + 562(Pa) \tag{2}$$

Where "R" is rainfall erosivity value (MJ mm ha$^{-1}$h$^{-1}$yr$^{-1}$) and "Pa" is mean annual rainfall (mm). For this purpose, nine years mean annual rainfall data was obtained from CHIRPS website.

As depicted in Figure 3, rainfall is higher in the lowland parts of the sub-basin, because in the highland region the rainfall characteristic is shower. Thus, the lower part of the sub-basin has higher erosivity values than middle and upper parts. The spatial distribution of the rainfall erosivity factor of north Gojjam sub-basin ranged from 657.34 to 921.73 MJ mm ha$^{-1}$h$^{-1}$yr$^{-1}$ (Figure 3).

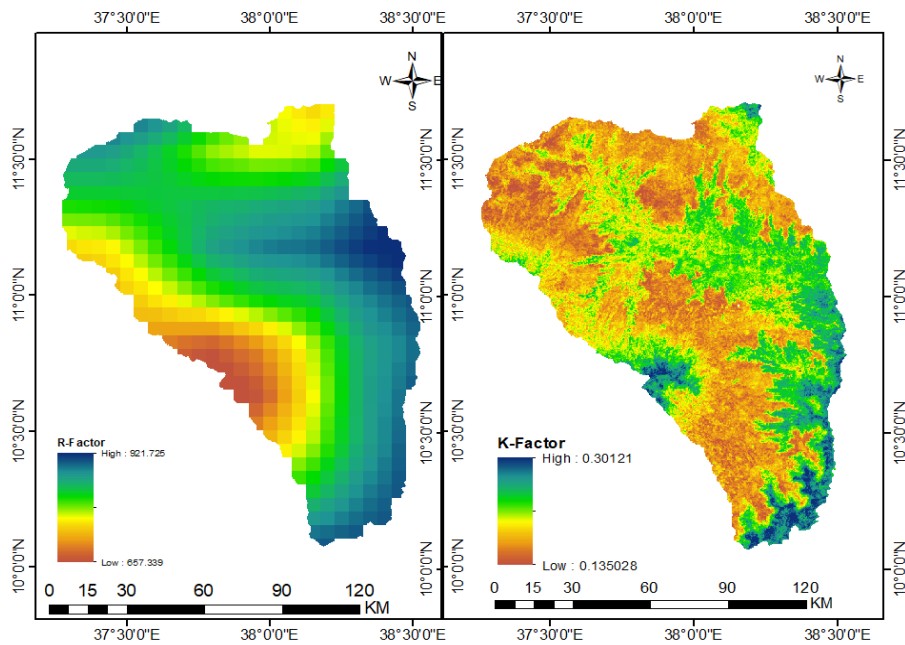

**Figure 3.** Rainfall erosivity (left) and soil erodibility (right) in the north Gojjam sub-basin.

Soil erodibility factor (K-factor) represents the sensitivity of soil particles and surface materials to detachment and transport through the power of rainfall and runoff [83,84]. It is associated with soil characteristics such as organic matter, texture, structure, permeability, and total stability [85]. The availability of organic matter in the soil decreases soil loss because it makes soil particles attached together [85]. There are different formulas developed by scholars to determine soil erodibility status [86]. Among others, Equation (3), shown below, is a formula widely used to calculate soil erodibility in the different environments. Using this approach, the K-factor of the study area was calculated using soil properties, or texture class and organic matter content [87].

$$K = \left\{ 0.2 + 0.3\exp\left[ 0.0256\text{SAN}\left( 1.0 - \frac{\text{SIT}}{100} \right) \right] \right\} \left( \frac{\text{SIT}}{\text{CLY} + \text{SIT}} \right)^{0.3} \left( 1.0 - \frac{0.25\text{C}}{\text{C} + \exp(3.72 - 2.95\text{C})} \right) \left( 1.0 \right.$$

$$\left. - \frac{0.7\left( \frac{1-\text{SAN}}{100} \right)}{\left( \frac{1-\text{SAN}}{100} \right) + \exp\left( -5.51 + 22.9 \left( \frac{1-\text{SAN}}{100} \right) \right)} \right) \tag{3}$$

where SAN = sand in %, SIT = silt in %, CLY = clay in %, and C = organic carbon in %.

The result indicated that soil erodibility value in the north Gojjam sub-basin was ranged from 0.13 t/ha/MJ/mm on upper and lower to 0.30 t/ha/MJ/mm on the middle part (Figure 3).

Topographic factor (LS-factor) refers the effect of topography on soil erosion. Slope length (L) and slope gradient (S) factors are joined in a single index, LS-factor, to describe the topographic factor for soil loss [88]. The slope length refers to the distance from the point of origin of overland flow to the point where either the slope gradient declines enough in which sedimentation starts or the runoff water enters a well-defined channel [80,89]. Soil loss increases with increases in slope gradient (S) and slope length (L) resulting from respective increases in velocity and volume of surface runoff [30]. We used modified equations for computing the topographic factor (LS-factor) suggested by Renard et al. [90]:

$$L = \left( \frac{\lambda}{22.13} \right)^m \tag{4}$$

$$m = {}^F\!/\!{}_{(1 + F)} \tag{5}$$

$$F = \frac{\sin\beta / 0.896}{3.0(\sin\beta)^{0.8} + 0.56} \tag{6}$$

where L is a slope length factor, $\lambda$ is the multiplication of flow accumulation and cell size, m is slope length exponent, F is calculated for conditions when the soil is moderately susceptible to both rill and inter-rill erosion, while $\beta$ is the slope angle in degree (slope in degree × 0.01745).

$$S = 10.8 \times \sin\beta + 0.03 \quad \delta < 9\%$$

$$S = 16.8 \times \sin\beta - 0.05 \quad \delta \geq 9\% \tag{7}$$

where S is a slope steepness factor, $\beta$ is the slope angle in degree, and $\delta$ is slope gradient in percent. The LS-factor map was generated from a 30 m pixel resolution ASTER global digital elevation model (GDEM) [46]. After calculating L and S values the LS value was computed by multiplying the two together. As depicted in Figure 4, the LS-factor value in the study sub-basin varied from 2 to 109, but the majority of the sub-basin has LS value of less than 10 (Figure 4).

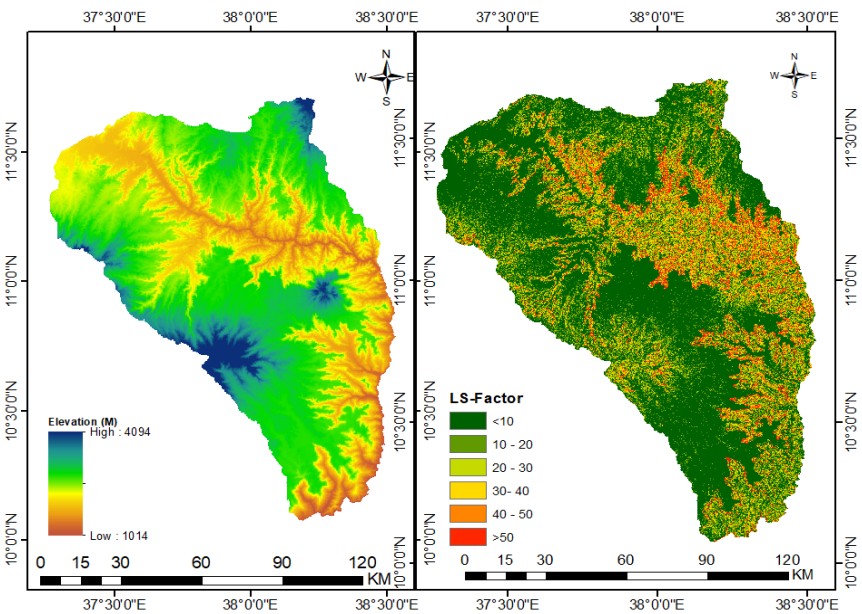

**Figure 4.** Digital elevation model (left) and topographic factor (LS-factor) (right) maps of the north Gojjam sub-basin.

Cover management factor (C-factor) describes the effect of cropping and management practices in agricultural land and ground cover (i.e., grass and tree canopy) on non-agricultural land on reducing soil loss [26,30]. To compute cover factor values in the RUSLE model, data are required associated with the role of plant canopy, crop residues, soil management practice, surface roughness, and moisture content of soil [80,83]. Acquiring each of the parameters is difficult due to the scarcity of data for combination and calibration. Alternatively, the C-factor can be calibrated from the LULC map. Using this simpler approach, the C-factor LULC map of the study area was prepared from a Landsat 8 image acquired in January 2017 using supervised classification techniques in ERDAS IMAGINE14 [91] (Figure 5). Ground truth data were used for reference for classification and accuracy assessment validation. The overall classification accuracy was 92% and the kappa coefficient index was 0.9. After classification was performed, the raster map was converted to vector format to assign C-factor values for each LULC type based on suggestions in previous literature (Table 1). Finally, the C-factor map changed to the raster layer using the raster conversion tool in ArcGIS10.5. This method was employed by different authors in studies of the Ethiopian highlands [26,30,92,93]. The cover management factor value in the sub-basin varied from 0 on water bodies to 0.6 on bare land (Figure 5). The smallest value indicates lower susceptibility to soil erosion.

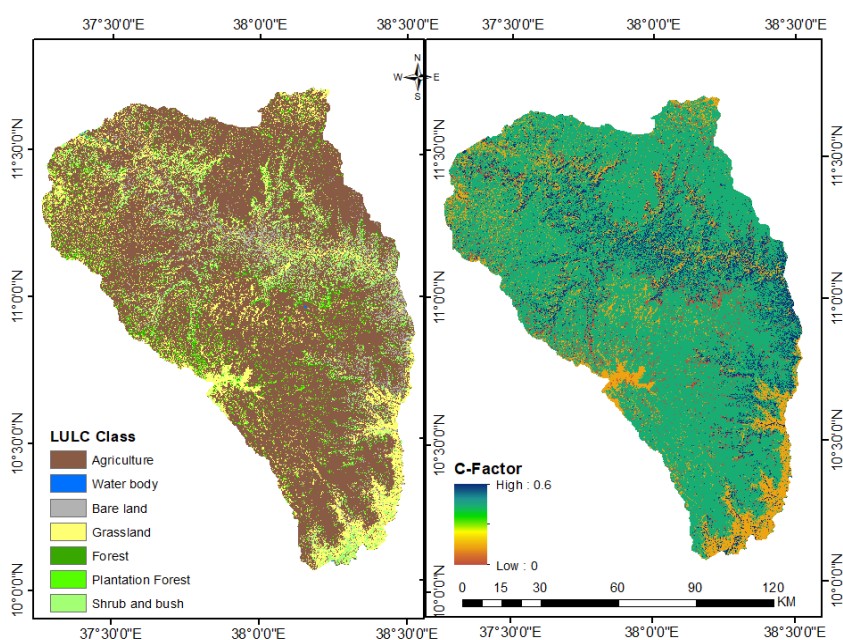

**Figure 5.** Land use and cover types (left) and the cover factor (right) values in the north Gojjam sub-basin.

**Table 1.** Adopted cover management factor (C-factor) values for different land uses/cover types.

| LULC | C-Value | Sources |
|---|---|---|
| Natural Forest | 0.01 | [30,94] |
| Plantation Forest | 0.01 | [30,94] |
| Shrub and bush | 0.20 | [30,94] |
| Grassland | 0.05 | [82,94] |
| Agriculture | 0.15 | [82,94] |
| Bare land | 0.60 | [94] |
| Waterbody | 0.00 | [26] |

Erosion management practice (P-factor) is a measure of the effect of soil management to reduce the extent of soil loss [88,95]. It involves several types of agricultural land management practices, such as terracing, contour farming, and strip cropping [81]. Unlike the previous studies in Ethiopia [26,30,94,96], this study generated P-values from conservation technology instead of using land use and topography. To determine agricultural land management practices for this study, intensive field observation, key informant interviews, and Google Earth Pro [49] observations were employed. Soil bund, stone bunds, hillside terraces, traditional ditches, cutoff-drains, waterways, check-dams and plantation on bund, afforestation, and revegetation have been implemented to various extents. From these technologies, we considered only soil and stone bunds conservation structures because most other Sustainable land management (SLM) options observed in the sub-basin were not well-designed and not widely used. As Figure 6 and Table 2 shows, almost half of the study areas were well terraced (48%), mainly the upslope parts, while about 30% of landscape needs structural measures and 17% of the sub-basin shows no need for conservation practices. The estimated *p*-values ranged from 0.02 on the terraced area to 1 on non-terraced agricultural land use types. The smaller value shows less vulnerability to soil loss (Table 2 and Figure 6).

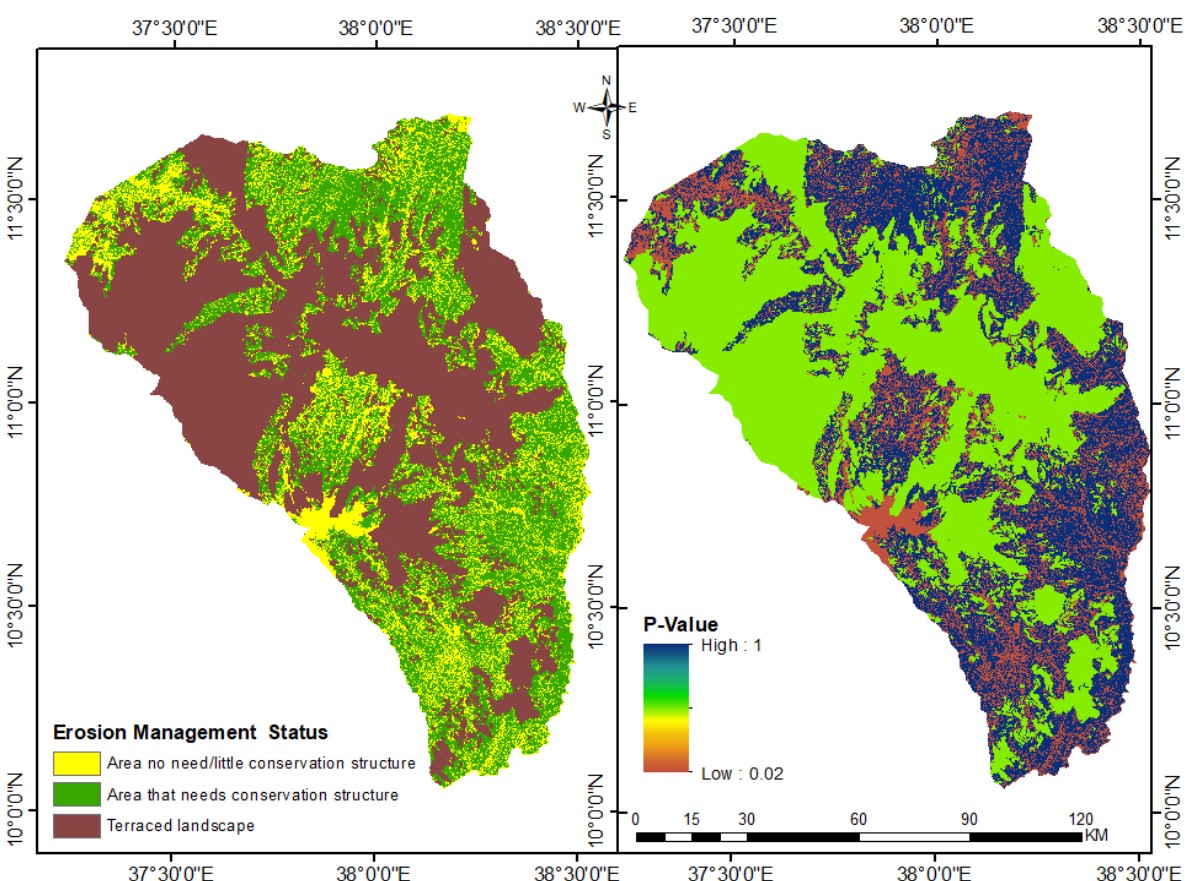

**Figure 6.** Erosion control measures (left) and management (P-factor) values (right).

**Table 2.** Erosion control status and management factor (P) values.

| Erosion Control Measures | Area in Percent | P-Factor | Sources |
|---|---|---|---|
| Area no need/little conservation structure | 17.79 | 0.02 | [21] |
| Area that need conservation stricture | 33.90 | 1.00 | [21] |
| Terraced landscape | 48.31 | 0.50 | [63] |

Finally, the annual soil loss was estimated on a cell-by-cell basis of multiplying the five RUSLE factors using Equation (1). As Landsat images and ASTER-GDEM [46] used in this study had 30 m spatial resolution, all the raster maps were resampled to 30 × 30 m cell size and re-projected to UTM Zone 37° N, WGS 1984 datum. The estimated annual soil loss was classified into six severity categories following soil erosion severity classification standards suggested by Haregeweyn et al. and Yesuph et al. [26,97].

The validation and consistency of the model output was compared with the quantitative outputs of previous experimental observations and similar empirical studies conducted in Ethiopia, mainly in the northwestern highlands. In addition, selected field observations were carried out. In supporting this process, the color printed model output soil erosion severity map was taken in the field to check the reality on the ground. Figure 7.

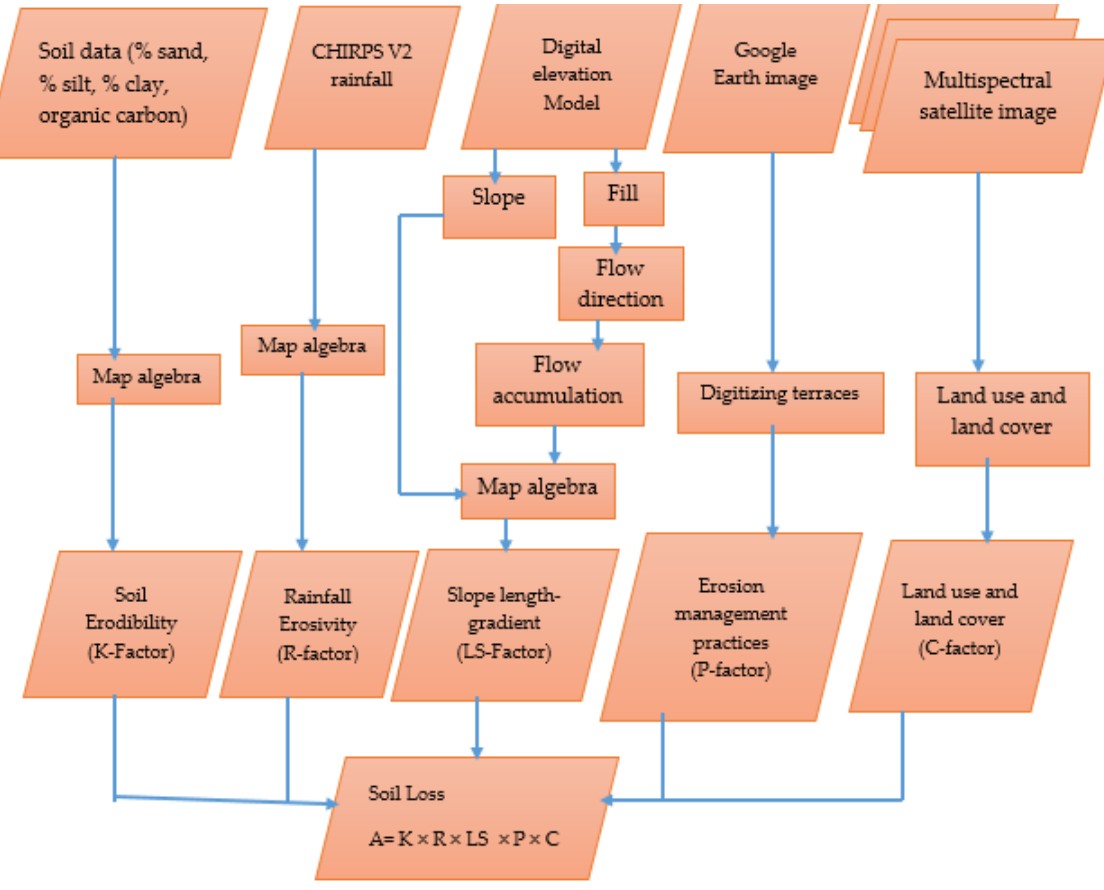

**Figure 7.** Flow chart showing the methodology of soil loss estimation.

Soil Compaction

Soil compaction is measured using soil bulk density [98]. Soil bulk density reflects soil compactness and soil health [55]. We obtained a soil bulk density map from Hengl et al. [48] in raster format and then reclassified based on the value range given in Table 3 to assign a level of compaction that is used as a physical land degradation indicator.

**Table 3.** Soil compaction class [71].

| Soil Bulk Density Class | Compaction Status |
|---|---|
| <1 g/cm$^3$ | Low soil compaction |
| 1–1.25 g/cm$^3$ | Medium soil compaction |
| 1.25–1.55 g/cm$^3$ | High soil compaction |
| >1.55 g/cm$^3$ | Very high soil compaction |

Soil Drainage

Soil drainage refers to the rate at which excess water moves from the soil surface through the soil profile. Poor drainage causes waterlogging in wet season and crusting in dry season [99]. Drainage class can be determined from observation of water table, soil wetness, landscape position, and soil morphology. According to the USDA [72], soil drainage can can be classified into seven classes based on the rate of water removal from the soil. To determine soil drainage status, a soil drainage raster map was acquired from Hengl et al. [48] and reclassified based on the values given in Table 4.

**Table 4.** Soil drainage classes status [72].

| Drainage Class | Level Drainage | Status Description |
|:---:|:---:|:---:|
| 1 | Very poor | Excessively drained |
| 2 | Poor | Somewhat excessively drained |
| 3 | Imperfect | Well drained |
| 4 | Moderate | Moderately well drained |
| 5 | Well | Somewhat poorly drained |
| 6 | Somewhat excessive | Poorly drained |
| 7 | Excessive | Very poorly drained |

Soil Depth

Soil thickness has a relation to soil quality and crop productivity [75]. Soil in which the rooting depth is limited by the presence of a physical constraint is generally less productive. Deep soils are favorable for the growth and development of plant roots with a higher supply of nutrients and minerals [75], implying low degradation level. To estimate soil depth and use it for the indicator of physical land quality status in our study area, the raster soil depth map was obtained from Hengl et al. [48] and reclassified into the categories listed in Table 5.

**Table 5.** The status of soil depth categories [73].

| Soil Depth Class | Severity Level | Description |
|:---:|:---:|:---:|
| <30 cm | Very low | Shallow soil |
| 30–50 cm | Low | Moderate shallow soil |
| 50–100 cm | Moderate | Deep shallow soil |
| 100–150 cm | High | Very deep shallow soil |
| >150 cm | Very high | Shallow soil |

2.3.2. Develop Biological Land Degradation Indicators

There are different indicators for assessing biological land degradation. Among others, vegetation cover, soil organic matter, and the reduction of soil organisms and soil fauna are common indicators of land degradation [100]. In this study, vegetation cover and soil organic matter were considered to estimate biological land degradation status.

Vegetation Cover

It is widely used to estimate land degradation status related to proxy indicators of greenness, vegetation density, vegetation growth, and biomass productivity [67]. We use the Soil Adjusted Vegetation Index (SAVI), a commonly applied remotely sensed index that offers an estimate of vegetation cover, and that overcomes limitations known to exist in the normalized vegetation difference index (NDVI) in areas with significant bare soil [101]. We implement SAVI following Chabrillat [67]:

$$\text{SAVI} = \frac{(\text{NIR} - \text{RED})}{(\text{NIR} + \text{RED} + \text{L})}(1 + \text{L}) \tag{8}$$

where NIR is the reflectance in the near-infrared and RED is the reflectance radiated in the visible red. L is a coefficient of vegetation density (ranging from 0 for very dense vegetation cover to 1 for very sparse). We tested several values for L, following recommendations from Huete [100] (0.25, 0.5, and 0.75). Visual interpretation of images led us to choose 0.5, which is an intermediate value that has been applied in several previous studies [100,102]. As a result, we used 0.5 as constant to determine vegetation index in this study.

Soil Organic Matter (SOM)

Soil organic matter is another useful biological land quality indicator [67]. Indeed, soil organic carbon together with soil pH has been recommended as a simple and reliable indicator of soil health and quality [67]. The amount of soil organic carbon is usually measured in the laboratory. However, it is difficult for this study to measure soil organic matter at the sub-basin level because of time and financial constraint. Consequently, a soil organic carbon raster map was obtained from Hengl et al. [48] and reclassified (Table 6) to determine the state of soil organic matter. It is estimated that soil organic matter contains 58% organic carbon. The soil organic carbon raster map was converted to soil organic matter following the equation used by Combs et al. [103].

$$Percentage\ of\ organic\ matter\ =\ Percentage\ of\ total\ organic\ carbon\ \times\ 1.72 \tag{9}$$

**Table 6.** Classes of soil organic matter status in soil [74].

| Category in % | Description |
|---|---|
| <0.2 | Very poor soil organic matter content in the soil |
| 0.2–0.6 | Poor soil organic matter content in soil |
| 0.6–1.2 | Medium soil organic matter contentin soil |
| 1.2–2.0 | High soil organic matter contentin the soil |
| >2.0 | Very high soil organic matter contentin the soil |

*2.4. Chemical Land Degradation Indicators*

Soil chemical degradation refers to undesirable changes in soil chemical characteristics, driven primarily by human intervention [104]. The most important indicators of chemical land degradation are soil acidity, salinity, and sodicity [104]. In this study we consider only soil acidity since the study area is humid and the application of chemical fertilizers and removable of crop residue are common. Soil acidity can be measured using pH value from solution soil in water [69]. To identify the status of soil acidity in the sub-basin, the soil pH raster map developed by Hengl et al. [48] was used. Then, the soil acidity digital map was classified using standardized soil pH classes (Table 7).

**Table 7.** Category of soil acidity level based on pH value [70].

| pH Value | Description |
|---|---|
| <4.5 | Extremely acid soils include acid sulfate soils |
| 4.5–5.5 | Very acid soils suffering often from toxicity |
| 5.5–7.2 | Acid to neutral soils: these are the best pH conditions for nutrient availability and suitable for most crops |
| 7.2–8.5 | These pH values are indicative of carbonate-rich soils |
| >8.5 | Indicates alkaline soils often very alkaline soils |

## 3. Results and Discussion

*3.1. Physical Land Degradation Indicators*

**Annual soil loss:** RUSLE model results indicate that much of the middle part of the sub-basin has lost topsoil at a rate of 0 to 75 t ha$^{-1}$yr$^{-1}$, and that soil loss rates exceed 75 t ha$^{-1}$yr$^{-1}$ in upstream and downstream zones as well as in some erosion hotspot areas (Figure 8). As indicated in Table 8, about 31.3% and 19.3% of the sub-basin experienced a very low and low soil loss rate, ranging from 0–5 t ha$^{-1}$yr$^{-1}$ and 5–15 t ha$^{-1}$yr$^{-1}$, respectively. The result shows that about 13.6% of the sub-basin experienced soil loss ranging from 15 to 30 t ha$^{-1}$yr$^{-1}$, which is characterized as a moderate erosion rate. Further, about 17.3% of the sub-basin lost topsoil with rates from 30 to 75 t ha$^{-1}$yr$^{-1}$, indicating high to very high soil loss rate. The remaining 18.5% of the sub-basin was under severe erosion rate with soil

loss exceeding 75 t ha⁻¹yr⁻¹. As shown in Table 8, the area under high to severe soil loss class covers about 35.9% area of the sub-basin, found in most upper and lower parts in very steep sloped areas (Figure 8). The average annual soil loss of the entire north Gojjam sub-basin was estimated at 46 t ha⁻¹yr⁻¹. This implies that the average soil loss in the study area is greater than two-times of the maximum (18 t ha⁻¹yr⁻¹) soil loss tolerance value given by Hurni [81] for the Ethiopian highlands. It implies that a total of 65.2 million tons of soil has been lost annually from the entire sub-basin. Any soil loss rate greater than 10 t ha⁻¹yr⁻¹ will not be restored in a period of 5 to 10 decades [105]. Accordingly, nearly half of the north Gojjam sub-basin was beyond the threshold of soil loss tolerance level (Table 8).

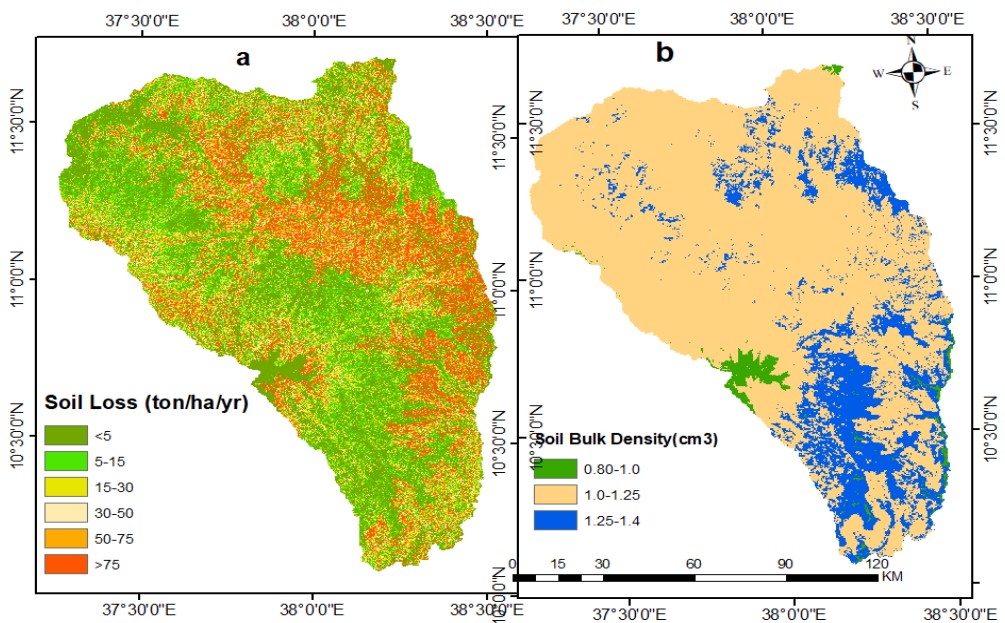

**Figure 8.** The annual soil loss (**a**) and soil bulk density (**b**) of the north Gojjam sub-basin.

Consistency and validation of the model estimation: The estimated average soil loss rate and the spatial patterns of this study are, in general, accurate, compared to what can be observed in the field as well as findings from previous experimental studies. Based on field assessment of rill and inter-rill erosion, Bewket and Sterk [106] found the annual soil loss ranged from 18 to 79 t ha⁻¹yr⁻¹ in parts of the same and adjacent watershed of this study sub-basin. Similarly, in five years of monitoring in an experimental micro-watershed (the Anjeni) located to the northwest adjacent to this study area, soil loss from cultivated fields under the traditional land-use practices was ranged from 17 to 176 t ha⁻¹yr⁻¹ [107]. Recently, Belayneh et al. [108] confirmed that the mean rate of soil loss in the new- and old-graded soil bund-treated and non-treated plots was 23.5, 45.6, and 58.1 t ha⁻¹yr⁻¹, respectively using experimental study from cultivated land in Gumara sub-watershed located in the present study sub-basin. Hurni [109] in Ethiopian highland estimated average soil loss from cultivated fields at 42 t ha⁻¹yr⁻¹ accounting from re-deposition of mobilized sedimentation. In addition, to check the validity, selected field observations were carried out. In supporting this process, the color-printed model output soil erosion severity map was taken into the field and checked against the reality on the ground.

The estimated soil loss rate of this study was also consistent with the empirical evidence from previous published studies. For example, 24.3 t ha⁻¹yr⁻¹ average soil loss was reported in the Gelana sub-watershed, north Wollo [96]; about 23.7 t ha⁻¹yr⁻¹ attested in the Geleda watershed [110]; nearly 24.9 t ha⁻¹yr⁻¹ reported in the Enfraz watershed [94]; the study conducted soil loss from the entire upper Blue Nile basin confirmed about 27.5

t ha$^{-1}$yr$^{-1}$ [26]; other results reported a relatively comparable estimation result to the present study from Jabi Tehinan district (30.6 t ha$^{-1}$yr$^{-1}$), west Gojjam zone [89]. The result of this study was most similar to recent empirical studies. For example, average soil loss of about 47.4 t ha$^{-1}$yr$^{-1}$and 42 t ha$^{-1}$yr$^{-1}$ was reported in the Koga watershed, upper Blue Nile [21,92]; about 49 t ha$^{-1}$yr$^{-1}$ mean soil loss was found in the Dembecha District, west Gojjam [93]. In contrast, some other studies undertaken in various regions of Ethiopia reported relatively higher average soil loss rate than this study. For instance, it was much lower than the mean soil loss rate of 243 t ha$^{-1}$yr$^{-1}$ reported by Zeleke [111] for north-western Ethiopian highlands due to rugged topography, low vegetation cover, and absence of land management technology; about 93 t ha$^{-1}$yr$^{-1}$ average soil loss was reported in the Chemoga watershed [30]; 84 t ha$^{-1}$yr$^{-1}$ average soil loss in northwestern Ethiopia found by Selassie Belay [112]; and 75 t ha$^{-1}$yr$^{-1}$ in the entire upper Blue Nile Basin [113].

The above empirical results indicated that though soil erosion is a common problem in Ethiopian highlands, the quantitative results are still uncertain and inconsistent. The variation may be observed from the heterogeneity of soil erosion determinants such as rainfall, soil property, topography, land management, and land use types, and may also stem from methodological differences between studies.

**Table 8.** Annual soil loss class and risk levels in the north Gojjam sub-basin.

| Soil Loss (t/ha/yr.) | Area (ha) | Percentage | Severity Level | Assigned Value | Risk Level |
|---|---|---|---|---|---|
| <5 | 447872.54 | 31.29 | Very slight | 1 | Very low |
| 5–15 | 276252.48 | 19.30 | Slight | 2 | Low |
| 15–30 | 193949.28 | 13.55 | Moderate | 3 | Medium |
| 30–50 | 141847.78 | 9.91 | High | 4 | High |
| 50–75 | 106350.05 | 7.43 | Very high | 5 | Very high |
| >75 | 265087.87 | 18.52 | Sever | 5 | Very high |

Soil compaction: The spatial distribution of soil compaction in the sub-basin ranged from 0.8 to 1.4 g/cm$^3$ (Figure 8). Table 9 indicates that 80.9% of the sub-basin soil bulk density was between 1 and 1.25 g/cm$^3$, indicative of medium compaction. The result shows that the status of soil compaction in a large area of the sub-basin can be considered as medium, which is attributed to the absence of heavy machine farming activity in the area. Low soil compaction was found in colder climate zones and areas with and more vegetation cover, located in a mountain area where higher precipitation rate and low temperature are found. The result implies that soil compaction was not much of a problem in the north Gojjam sub-basin.

Soil drainage: As seen from Table 9, about 74.4% of sub-basin has good drainage characteristics. The spatial distribution of soil drainage is shown in Figure 9. The result is similar to the soil drainage map prepared by the agricultural transformation agency [114] in the Amhara regional state and local land users' perception.

Soil depth: Spatial distribution of soil depth in the sub-basin ranged from 25–175 cm (Figure 9). As presented in Table 9, about 40.9% of the sub-basin has very deep soils (>150 cm), which indicates a very low land degradation level with respect to this indicator, while about 50.3% of the sub-basin has shallow soil, ranging from 25–30 cm, which indicates a very high degree of degradation. Most midland parts of the sub-basin are characterized by high soil depth while the lower part has very shallow soil, reflecting higher erosion rates and greater soil degradation.

**Table 9.** Statistics for physical land degradation indicators in the north Gojjam sub-basin.

| Factor | Classes | Area (Ha) | Percentage | Assigned Value | Degradation Level |
|---|---|---|---|---|---|
| Soil bulk density (g/cm³) | <1 | 19,323.36 | 1.35 | 2 | Low |
| | 1–1.25 | 1,158,399.65 | 80.93 | 3 | Moderate |
| | 1.25–1.55 | 253,636.99 | 17.72 | 4 | High |
| Level of drainage | Poor | 2147.04 | 0.15 | 1 | Very low |
| | Imperfect | 110,787.26 | 7.74 | 2 | Low |
| | Moderate | 214,990.27 | 15.02 | 3 | Moderate |
| | Well | 1,064,788.70 | 74.39 | 4 | High |
| | Somewhat excessive | 38,646.72 | 2.70 | 5 | Very high |
| Soil depth class (cm) | 25–30 | 719,544.67 | 50.27 | 5 | Very high |
| | 30–50 | 3864.67 | 0.27 | 4 | High |
| | 50–100 | 1145.09 | 0.08 | 3 | Moderate |
| | 100–150 | 122,524.42 | 8.56 | 2 | Low |
| | >150 | 584,281.15 | 40.82 | 1 | Very low |

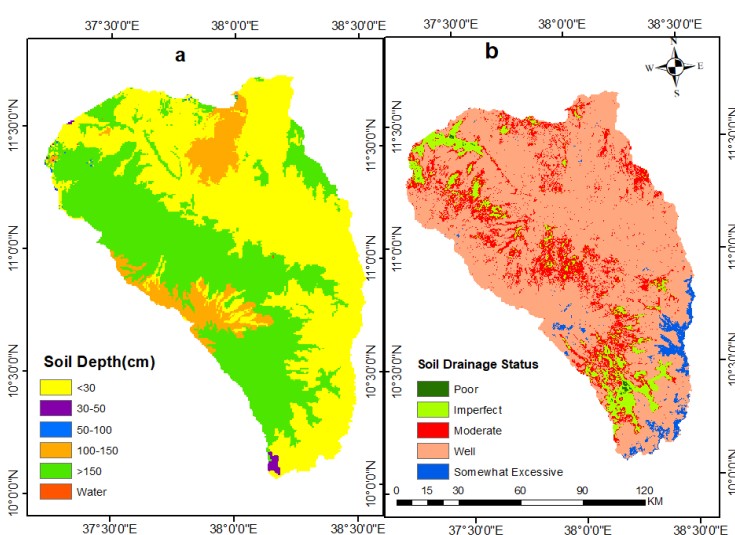

**Figure 9.** Soil bulk depth (**a**) and soil drainage (**b**) in the north Gojjam sub-basin.

### 3.2. The State of Physical Land Degradation

The weights of the four indicators which contribute for physical land degradation (soil erosion, soil compaction, soil drainage, and soil depth) have been derived through a pairwise comparison. The weight has been given based on the influence of every subclass for land degradation. The calculated pairwise comparison matrix consistency ratio is 0.01, indicating a consistent comparison. As the overlay analysis result depicted in Figure 10 shows, the majority (72.8%) of the sub-basin physical land degradation level was moderate. The weights of pairwise comparison matrix result shows that soil drainage, soil depth, soil erosion, and soil compaction were the most to the least important physical land degradation indicators in the north Gojjam sub-basin (Table 10). This implies that the wider area of the sub-basin's physical land degradation status was moderate (Figure 10). Most low vegetation cover areas fall under the shallow soil depth due to the presence of high soil erosion.

**Table 10.** Pairwise comparison matrix for physical land degradation indicators.

| Criteria | Soil Drainage | Soil Depth | Soil Erosion | Soil Compaction | Criteria Weighting |
|---|---|---|---|---|---|
| Soil drainage | 1 | 2 | 3 | 5 | 45 |
| Soil depth | 0.5 | 1 | 2 | 3 | 27 |
| Soil Erosion | 0.33 | 0.50 | 1 | 3 | 20 |
| Soil compaction | 0.2 | 0.33 | 0.33 | 1 | 8 |

Similarly, local farmers in the study area reported physical land degradation problems in the form of low soil infiltration rate, soil depth reduction, and soil erosion. In both formal and informal discussions, farmers explained that there is the increasing problem related to soil compaction on their farm field. They did report that due to soil compaction, rainwater infiltration into the soil has been decreasing and result in increasing soil erosion rate. Soil depth has decreased due to erosion from runoff water and continuous cultivation, particularly on steep slopes and croplands. These discussion points suggest that physical land degradation may be an increasing problem in the sub-basin. According to Amede [115], soil erosion by water is the main land degradation agent in the Amhara regional state.

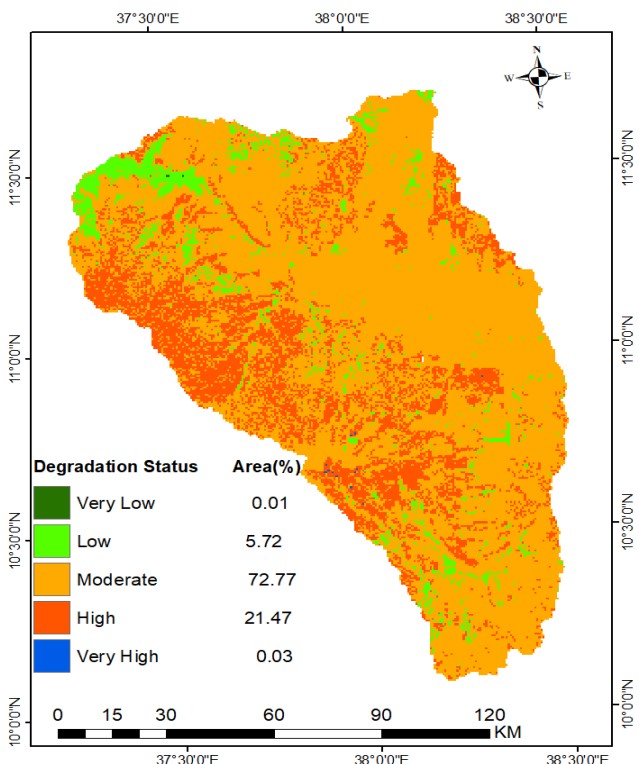

**Figure 10.** State of physical land degradation in the north Gojjam sub-basin.

### 3.3. Biological Land Degradation Indicators

Vegetation cover: As Figure 11 shows, the spatial patterns of vegetation indices in the north Gojjam sub-basin ranged from −0.2 to 0.86. As seen in Table 11, about 20.9% and 60.3% part of the sub-basin has moderate and poor vegetation cover, respectively. These areas are characterized as moderate and high land degradation status, respectively. As shown in Table 11 and Figure 11, more than half of the sub-basin was classified as having high to very high land degradation level according to the vegetation indicator. The severity was higher in the lowland area than the highland in the sub-basin, where the concentrations of plantation and grazing land were low.

**Table 11.** Statistics of mean soil adjusted vegetation index (SAVI).

| SAVI Classes | Area (ha) | Area (%) | Cover Status | Assigned Values | Degradation Level |
|---|---|---|---|---|---|
| <0.1 | 67,972.77 | 4.75 | Very poor | 5 | Very high |
| 0.1–0.2 | 862,529.67 | 60.26 | Poor | 4 | High |
| 0.2–0.3 | 298,752.48 | 20.87 | Moderate | 3 | Moderate |
| 0.3–0.4 | 172,708.74 | 12.07 | High | 2 | Low |
| >0.4 | 29,396.34 | 2.05 | Very high | 1 | Very low |

The status of soil organic matter (SOM): The spatial distribution of soil organic matter content in the north Gojjam sub-basin ranged from 0.15% to 1.86% (Figure 11). As Table 12 depicts, for a large area (72.6%) of the sub-basin soil organic matter proportion ranged from 0.2 to 0.6%, a low SOM value that is considered to be highly degraded for this indicator. Low SOM is associated with greater erosion and with low levels of micro-organisms in the soil. Higher SOM indicates the presence of richer flora and fauna residues at various stages of decomposition, and soils that are rich in humus content. The low level of SOM found over most of the north Gojjam sub-basin may be due to continued cultivation and collection of crop residues for domestic energy, livestock feed, and home building materials in favor of use as mulch on the farm fields. Free grazing is also one of the causes of decreasing crop residue in the study sub-basin.

**Table 12.** Levels of soil organic matter of topsoil in the north Gojjam sub-basin.

| Category in % | Area (Ha) | Percentage (%) | Level of SOM | Severity Level | Assign Value |
|---|---|---|---|---|---|
| 0.15–0.2 | 19,624.79 | 1.73 | Very low | Very high | 5 |
| 0.2–0.6 | 348,429.04 | 72.56 | Low | High | 4 |
| 0.6–1.2 | 1,038,570.57 | 24.34 | Medium | Moderate | 3 |
| 1.2–1.86 | 24,735.61 | 1.37 | High | Low | 2 |

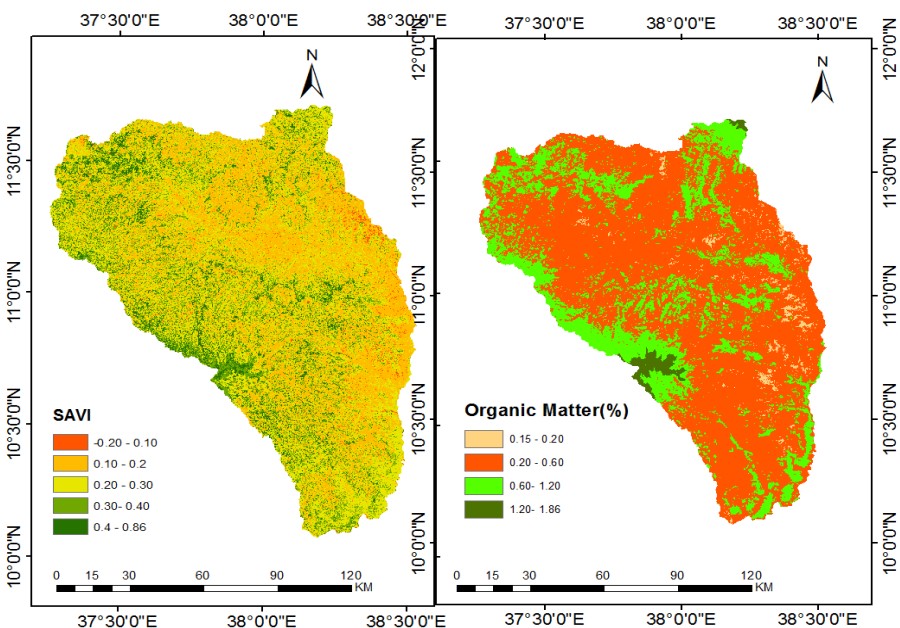

**Figure 11.** Soil Adjusted Vegetation Index (left) and soil organic matter (right) of the sub basin.

### 3.4. The Status of Biological Land Degradation

Similar to physical land degradation indicators, the weights of the two indicators which contribute for biological land degradation (vegetation cover and soil organic matter) have been derived through a pairwise comparison matrix. The weight has been given

based on the influence of every sub-class for land degradation. As presented in Table 12, soil organic matter was a more influential indicator than vegetation covers for biological land degradation in the sub-basin. The calculated pairwise comparison consistency ratio was zero, which implies that the comparison was perfectly consistent and the comparison is acceptable. As the weight overlay analysis shows in Figure 12, 60% and 4.7% of the sub-basin was highly to very highly degraded. About 30.8% and 4.4% of the sub-basin has been degraded in moderate and low-level risk. This suggests that more than half of the north Gojjam sub-basin was highly degraded biologically due to vegetation cover degradation and soil organic matter depletion.

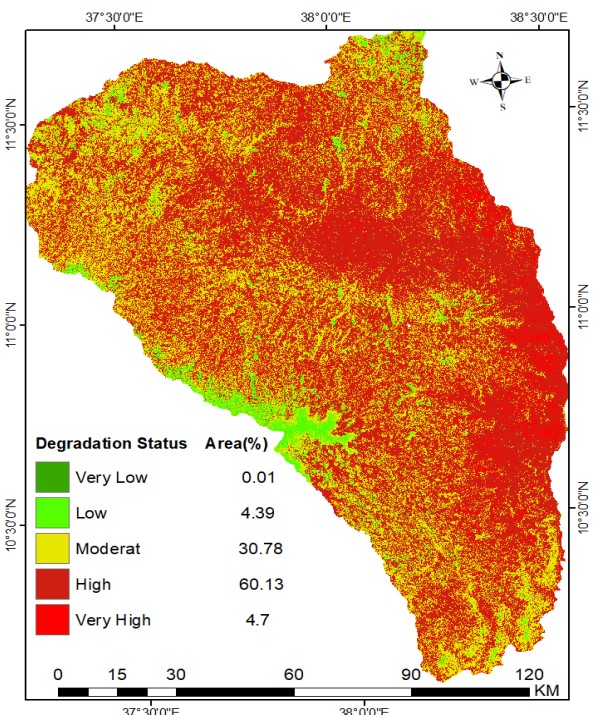

**Figure 12.** Biological land degradation status in the north Gojjam sub-basin.

This result is consistent with local land users' view. In the formal and informal discussions, farmers reported that both fauna and flora have been declining through time on their farmland due to over-exploitation of natural resources. According to the respondents' view, soil nutrient loss by water is an unsolved form of land degradation and is an ongoing problem. They conclude that the loss of nutrient and reduction of organic matter has increased over recent few decades in the sub-basin. Table 13.

**Table 13.** Pairwise comparison matrix of biophysical land degradation indicators.

| Criteria | Organic Matter | Vegetation Cover | Criteria Weighting |
|---|---|---|---|
| Organic matter | 1 | 2 | 66.7 |
| Vegetation cover | 0.5 | 1 | 33.3 |

### 3.5. The State of Chemical Land Degradation

The spatial variation of chemical land degradation in the form of soil acidity in the north Gojjam sub-basin varied from 5 to 7.8 pH value (Figure 13 and Table 14). The result shows that soil acidity level for about an area covered about 4% was less than 5.5 pH value which is considered as high. High soil pH concentration is found in the Choke mountain reserved area, where wet climate condition and water availability is higher as well as the area covered by natural forest and afro-alpine grass. Wet climate and high rainfall leach

soluble nutrients from soil, such as calcium and magnesium which are specifically replaced by aluminum and increased potential for acidic soils [116]. The decomposition of organic matter produces hydrogen ions, which are responsible for soil acidity formation [117].

About 39% part of the sub-basin experienced soil acidity ranged from 5.5 to 6.7 pH value, which expressed a high level of degradation. As Figure 14 observed most highland and midland parts of the sub-basin, in which continuous agriculture activity and continuous application of chemical fertilizers takes place, as well as eucalyptus plantation being common, were vulnerable to soil acidity. This might be from the application of acid-forming fertilizers and over-cultivation. According to local experts, due to population growth and persistent demand for food and fuel, the removal of agricultural by-products (crop residues) and continuous crop harvest and use of acid forming inorganic fertilizers make an important contribution to soil acidity development in the sub basin. Continuous application of chemical fertilizers with nitrogen and/or phosphorus nutrients only in the form of diammonium phosphate (DAP) and urea has adversely affected soil chemical properties [116]. Land used for eucalyptus fields are the most affected by soil acidity [116].

The majority of the area (55.8%) the pH of soils in the sub-basin varied from 6.7 to 7.3, a range that is considered neutral, whereas the pH value for the remaining 1.23% of the basin ranged from 7.3 to 7.8, which is characterized as alkaline soil. Low soil acidity level located in low land areas, in most dry areas. The result implies that almost half of the study area was vulnerable to chemical land degradation. However, no area in the sub-basin was affected by strong soil acidity. FGDs and key informant participants reported that the application of lime on the cropland has been increasing in the last 10 years, due to the increasing problem of soil acidity, mainly in the heavily cultivated middle and upper part of the sub-basin. Nevertheless, most farmers do not use lime to reclaim acid soil due to the scarcity of lime supply.

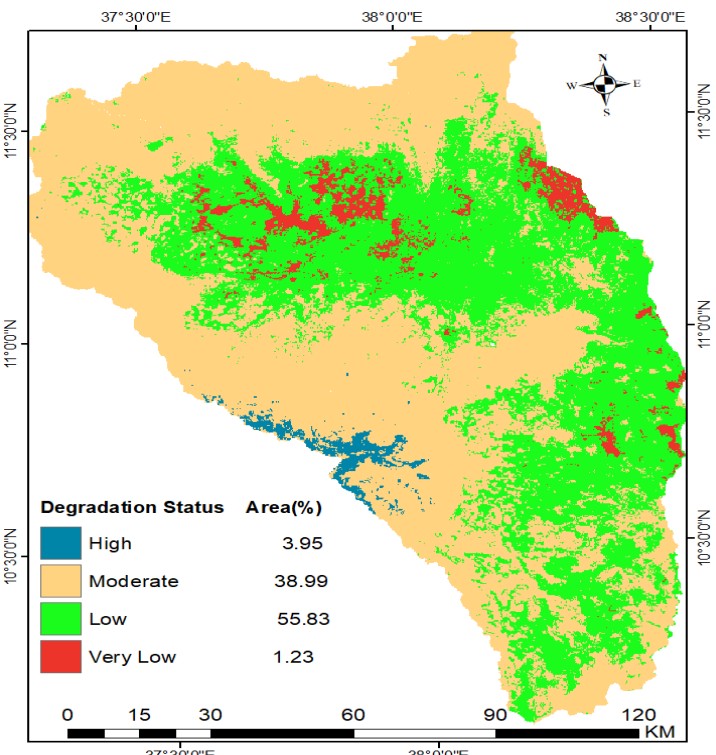

**Figure 13.** Soil acidity status in the north Gojjam sub-basin.

**Table 14.** The status of soil acidity in the north Gojjam sub-basin.

| Soil pH | Area (Ha) | Percentage (%) | Level | Assigned Value |
|---------|-----------|----------------|-------|----------------|
| 5–5.5 | 56,525.14 | 3.95 | High | 4 |
| 5.5–6.7 | 558,078.76 | 38.99 | Medium | 3 |
| 6.7–7.3 | 799,124.40 | 55.83 | Low | 2 |
| 7.3–7.8 | 17,631.70 | 1.23 | Very low | 1 |

*3.6. The Status of Comprehensive Land Degradation in the North Gojjam Sub-Basin*

The comprehensive land degradation map of this study was produced by combining biological, physical, and chemical land degradation indicators. All the parameter raster maps were resampled to 30 × 30 m cell size and re-projected to UTM Zone 37° N, WGS 1984 datum. As seen from the pairwise comparison matrix result in Table 15, biological, physical, and chemical degradation indicators were the most to the least important factors for contributing comprehensive land degradation in the north Gojjam sub-basin. The weighted comparison consistency ratio was 0.09, and thus, the comparison was acceptable as the value is less than 0.10.

The result shows that about 32% of the sub-basin area exhibits low-level degradation while about 35.4% is moderately and 30.5% is highly degraded (Figure 14). The result shows that the spatial distribution of land degradation in the sub-basin was uneven. As depicted in Figure 14, the most highly degraded areas are located in the lower part of the sub-basin. This is a result of a number of factors: steep slopes, poor land management and continued cultivation, rugged topography, population pressure, and erratic rainfall. The moderately degraded areas were located in the middle elevation portion of the sub-basins, where the area is characterized by plain topography and low vulnerability to soil erosion. These factors are confirmed by local land users. Local communities explained that a combination of soil erosion, low vegetation cover, low soil organic matter, and soil acidity contributed to land degradation in the north Gojjam sub-basin. Moreover, they reported that climate condition, poor agricultural activity, poor grazing, and poor quality of soils had contributed to soil erosion in particular and land degradation in general in the sub-basin. Overall, the combined degradation analysis shows that more than 60% of the sub-basin was moderately to highly degraded. This implies that land degradation is a serious environmental and economic problem in the north Gojjam sub-basin.

**Table 15.** Pairwise comparison matrix of land degradation status in the north Gojjam sub-basin.

| Criteria | Biophysical Degradation | Physical Degradation | Chemical Degradation | Criteria Weighting |
|----------|------------------------|---------------------|---------------------|---------------------|
| Biophysical degradation | 1 | 3 | 7 | 69 |
| Physical degradation | 0.33 | 1 | 2 | 21 |
| Chemical degradation | 0.14 | 0.5 | 1 | 10 |

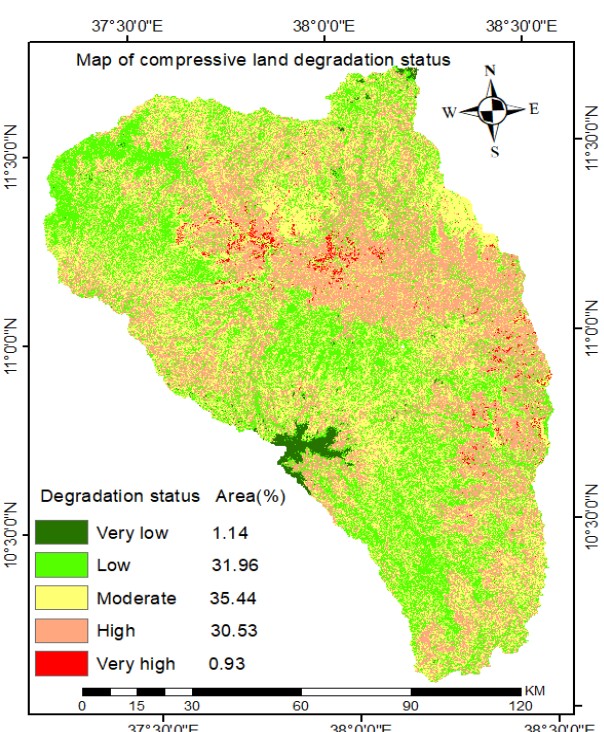

**Figure 14.** Comprehensive land degradation map of the north Gojjam sub-basin.

## 4. Conclusions and Policy Implications

Assessment of land degradation is prerequisite to develop evidence-based and efficient land management planning. To meet this need, this study mapped and quantified the status of comprehensive land degradation in a typical highland of Ethiopia, the north Gojjam sub-basin of the upper Blue Nile basin. To map and quantify different land degradation indicators into a single indicator, we followed the standardized classification technique in ArcGIS10.5 and the hierarchical spatial MCA approach. The rate of soil loss was estimated using the RUSLE model taking into account the basic factors of soil erosion, including topography, soil characteristics, rainfall, land cover, and local land management measures.

The RUSLE model yields an estimate that on average 46 t ha$^{-1}$yr$^{-1}$ or a total of 65.2 million tons of topsoil has been lost from the sub-basin annually. About 45.3% of the sub-basin was evaluated to be at high and very high soil loss risk. Most parts of the sub-basin suffered from high to very high biological land degradation. The majority of the sub-basin is moderately affected by soil acidity and physical deterioration of land quality, but biological land degradation was considered to be a more important factor in land degradation than chemical or physical indicators. The result of the combined land degradation indicators confirmed that more than 60% of the sub-basin was moderately to highly degraded.

The diverse aspects of land degradation in the sub-basin point to the need to integrate structural, biological, and agronomic land management measures to maintain sustainable environmental management and economic development. Particularly, lime application and organic fertilizer (compost, manure, and mulching) application are very important to reverse soil acidity and to improve soil fertility status. The adoption of agroforestry and economically viable multi-purpose perennial crops should be promoted to reverse soil degradation and reduce soil erosion in the sub-basin. This requires the collaboration of all the stakeholders to rehabilitate formerly degraded areas and to minimize the current degradation rate as well as to improve ecosystem health and maintain sustainable development in the area. Further, the study confirmed that the use of GIS and remote sensing technologies combined with the spatial MCA technique is a useful tool in mapping and

characterizing land degradation using a combination of spatial data indicators. This study considers rill and inter-rill erosion by water. Thus, future researchers should consider gully and river bank erosion. Soil quality can be measured using several soil essential elements such as hydrogen ion concentration, electrical conductivity, total nitrogen, available phosphorus, potassium, calcium, magnesium, sodium, and others. Future researchers should consider these gaps in the north Gojjam sub-basin. In general, land resources bases are dynamic and diverse across a region depending on the dominant socioeconomic and biophysical factors of that location. This emphasizes the need for regular engagement with farmers to address emerging opportunities and challenges.

**Author Contributions:** All authors have made substantial contributions equally to the design of the research work and the acquisition, analysis, or interpretation of data. All authors have read and agreed to the published version of the manuscript.

**Funding:** This research was funded by NILE-NEXUS: Opportunities for a sustainable food–energy–water future in the Blue Nile Mountains of Ethiopia. Belmont Forum. Addis Ababa University and John Hopkins University have also supported this research.

**Institutional Review Board Statement:** Informed consent was obtained from all subjects involved in the study.

**Informed Consent Statement:** Not applicable.

**Conflicts of Interest:** The authors declare no conflict of interest.

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
