# Peer review of "Mapping and Quantifying Comprehensive Land Degradation Status Using Spatial Multicriteria Evaluation Technique in the Headwaters Area of Upper Blue Nile River"

_sustainability, doi:10.3390/su13042244_

Round 1

Reviewer 1 Report

The paper is a serious contribution to literature. There are some issues that could be improved. The first aspect is related to the introductory part. The authors have not delved into the literature on land degradation in a broader context, whereby land is a natural resource. Land is a natural resource and production factor. Hence, I would just suggest the authors to refer to the recent literature, Sadik-Zada (2020) in Review of Development Economics and Sadik-Zada (2019) in The European Journal of Development Research, on the nexus between natural resources (here land) and economic development. Refer cursorily to Physiocracy in the introductory part. The paper is rather lengthy, but the length is justified against the backdrop of explanatory parts of the exposition
. I would, nevertheless, suggest that the authors shorten Abstract by at least 30 percent. Figure 2 is not overall readable. In the section 2.2 Data sources, you indicate the URLs. This must be done only in References. Mention only data sources and the respective year without indicating the full URL. The subsection 2.3.1 consists of just one sentence. It is too short. If so then merge 2.3.1 and 2.3.1.1 or extend 2.3.1. In the references sometimes the authors indicate the access date to the internet sources and sometimes not.

The authors made a good job. Nevertheless, they did not delve into the literature on natural resource abundance. Land is also a natural resource and production factor. This is a minor, not central issue. Hence, I would just suggest the authors to refer to the recent literature, Sadik-Zada (2020) and Sadik-Zada (2019), on the nexus between natural resources (here land) and economic development. Refer cursorily to Phisiocrats in the introductory part.

Author Response

Dear Sir/Madam,

Thank you so much for your critical reviews and comments. We found them very important and tried to address all the comments.

Respectfully,

Reviewer 2 Report

  1. Cut down explanation of AHP, which is a standard technique, and explain only its use to derive the weights.
  2. Fig 2 is not clear; some boxes are too small. 
  3. There are a large number of typos.
  4. Be consistent with notations, e.g. P for soil management (line 165), and not again for rainfall (line 172).
  5. Write equations clearly, especially exponents, subscripts, and divisions, e.g. Eqn 3 is confusing and SAD is not defined. Eqn 5, F/1+F should be F/(1+F).
  6. Line 202 "sin beta" should be beta.
  7. Line 285, use of 0.5 to estimate unknown vegetation cover is not satisfactory. There has got to be a better way. 
  8. Most important, the method relies substantially on the RUSLE equation and it is not empirically verified on the actual site. If so, you should provide an estimate of the variance.  

Author Response

Dear Sir/Madam,

Thank you so much for your comments and edits. We found them very valuable and tried to address them accordingly.

Respectfully,

Reviewer 3 Report

Mapping and quantifying land degradation status is very important for identifing vulnerable areas and to design sustainable landscape management. The proposed study shows how to map and quantify land degradation status in the north Gojjam sub-basin of the Upper Blue Nile River (Abbay) using GIS and Remote Sensing integrated with multicriteria analysis (MCA). The authors build compressive land degradation assessment, the study employed a combination of different biological, physical and chemical land degradation indicators. All indicators were standardized and weighted using analytical hierarchy and pairwise comparison techniques.
As the results, about 45.3% of the sub-basin was found to experience high to very high soil loss risk, with an average soil loss of 46 t ha-1 yr-1. More than half of the sub-basin was found to experience moderate to high level of biological degradation (low vegetation status and low soil organic matter level). 80.2% of the area is characterized with moderate physical land degradation level. Similarly, the status of chemical land degradation for about 55.8% and 40% of the sub-basin was grouped as low and moderate, respectively. The combined spatial multi-criteria analysis of biological, chemical, and physical land degradation indicators showed that about 1.14%, 32%, 35.4% and 30.5% of the sub-basin exhibited very low, low, moderate, and high degradation level, respectively.
The presented paper is good written scientific work. However some shortcomings should be identified before accepting. The list of comments is as follows:
1. The last paragraph in the Introduction should show the structure of the rest paper.
2. The contribution should be more emphases in the abstract and introduction section
3. Section 2.3 is not enough. There should be a short discussion about another popular MCDA methods, which can be used or not to solve this problem. The justification of used methods is very important. The brief overview of the current research directions is needed (and very important), e.g., are mcda methods benchmarkable? a comparative study of topsis, vikor, copras, and promethee ii methods; Identification of relevant criteria set in the mcda process—wind farm location case study; Identification of relevant criteria set in the mcda process—wind farm location case study; and similar
4. How do you prevent rank reversal phenomena, more information you can find, e.g., The rank reversals paradox in management decisions: The comparison of the ahp and comet methods
5. Figures 2 and 7 should be redraw
6. Lines 162 cdot should be used instead *
7. Standardize the font in models (1) - (7)
8. Which standardization formula do you use?
9. More information about future research directions should be added in the conclusion section.

Author Response

Dear Sir/Madam,

Thank you so much for your critical reviews and comments. We found them important and tried to address accordingly.

Respectfully,

Round 2

Reviewer 2 Report

1. There are still some typos, e.g. "As the results,..." in the Abstract.

2. Some figures and arrows are out of alignment and require fixing.

3. Equations should read as a sentence, e.g. We have

      y = 1+ 3G

    where y is xxx and G is xxx. Here "where" should be in lower case because it is part of the sentence.

4. On the variance of the estimates, empirical verification is not good enough because it is not rigorous and can be accidental.

Author Response

Dear Sir/Madam,

Thank you so much. We have addressed all the comments and believe that now, the article is in its final shape.

Regards,

Reviewer 3 Report

I suggest accepting in the current version

Author Response

Dear Sir/Madam,

Thank you so much for your commendable job in order to improve our article. We have made all the efforts to address all your comments.

Regards,

Belay simane
